# A Procedure for the Quantitative Comparison of Rainfall and DInSAR-Based Surface Displacement Time Series in Slow-Moving Landslides: A Case Study in Southern Italy

**Francesca Ardizzone** [1], **Stefano Luigi Gariano** [1,*], **Evelina Volpe** [1], **Loredana Antronico** [2], **Roberto Coscarelli** [2], **Michele Manunta** [3] **and Alessandro Cesare Mondini** [1]

1   CNR IRPI (Research Institute for Geo-Hydrological Protection, Italian National Research Council), Via Della Madonna Alta 126, 06128 Perugia, Italy
2   CNR IRPI (Research Institute for Geo-Hydrological Protection, Italian National Research Council), Via Cavour 4-6, 87036 Rende, Italy
3   CNR IREA (Institute for Electromagnetic Sensing of the Environment, Italian National Research Council), Via Diocleziano 328, 80124 Napoli, Italy
*   Correspondence: stefano.luigi.gariano@irpi.cnr.it; Tel.: +39-075-5014-424

**Abstract:** Earth observation data are useful to analyze the impact of climate-related variables on geomorphological processes. This work aims at evaluating the impact of rainfall on slow-moving landslides, by means of a quantitative procedure for identifying satellite-based displacement clusters, comparing them with rainfall series, and applying statistical tests to evaluate their relationships at the regional scale. The chosen study area is the Basento catchment in the Basilicata region (southern Italy). Rainfall series are gathered from rain gauges and are analyzed to evaluate the presence of temporal trends. Ground displacements are obtained by applying the P-SBAS (Parallel Small BAseline Subset) to three datasets of Sentinel-1 images: T146 ascending orbit, and T51 and T124 descending orbits, for the period 2015–2020. The displacement series of the pixels located in areas mapped as landslides by the Italian Landslide Inventory and sited within rain gauge influence regions (defined as 10 km circular buffers) are studied. Those displacement series are analyzed and compared to the rainfall series to search for correlations, by employing statistical and non-parametric tests. In particular, two landslides are selected and investigated in detail. Significant results were obtained for the T124 descending orbit for both landslides, for a 3-day cumulative rainfall and a 7-day delay of the slope response. Challenges in the whole procedure are highlighted and possible solutions to overcome the raised problems are proposed. Given the replicability of the proposed quantitative procedure it might be applied to any study area.

**Keywords:** Sentinel-1; statistical test; climate-related hazard; remote sensing; southern Italy

## 1. Introduction

Climate variables and their changes have effects on landslide activity and slope stability [1]. This is becoming even more evident under the effects of global warming and resulting climate change [2,3]. However, quantifying the type, extent, magnitude and direction of the climate changes on the slope stability remains difficult. This is mostly due to the fact that climate and landslides operate at different spatial and temporal scales and reconciling these differences is difficult and might produce uncertain results [4,5].

Differential Interferometric Synthetic Aperture Radar (DInSAR) allows identifying ground displacements over large areas, and studying the landslide activity at different time scales. Since the 1990s, the DInSAR technology has been used to measure the displacement of the earth's surface due to volcanic disturbances, seismic phenomena and subsidence [6]. Indeed, DInSAR allows the measurement of the deformation component along the radar line of sight (LOS) with large spatial coverage capability and accuracy of a fraction of the wavelength of the transmitted microwave signals (from a few to tens of cm). This

result is achieved by exploiting the phase difference (interferogram) between pairs of complex SAR images [7], usually referred to as Single Look Complex (SLC). Over the last decades, thanks to the availability of advanced SAR constellations (e.g., COSMO-SkyMed, TerraSAR-X/TanDEM-X, Sentinel-1) with improved capabilities in terms of revisit time, spatial resolution and ground coverage, DInSAR technology has been largely applied also to monitor over time surface deformations in several deformation scenarios, such as landslides, built-up environment, post-seismic phase and volcanic activity, as testified by the large number of related applications [8–12]. Indeed, the current SAR images are acquired with a spatial resolution of a few meters, a revisit time of some days and spatial coverage of hundreds of km.

In particular, the DInSAR technique is widely used and has broadly recognized tools for landslide mapping and monitoring due to the medium-to-high temporal and spatial resolution [13]. In their work, Solari and co-authors [13] provide an overview of data usage in Italy. The authors identified six classes. None of them is dedicated specifically to the relationship between the landslide activity and climate parameters (i.e., rainfall measurements). Recently, Moretto et al. [14], proposed a classification discerning five different levels of landslide predictability by SAR interferometry, listing: successful spatial and temporal prediction of the time of failure with SAR data ("predictable landslide"); observation of a worsening of the situation, without the ability to predict the time of failure ("critical behavior predictability"); detection of spatial anomaly allowing to accurately delimit the slope instability without temporal insight ("spatial predictability"); classification of the state of activity of a landslide based on the geological/geomorphological interpretation of SAR data ("qualitative spatial predictability"); displacements not observable by InSar ("unpredictable landslide"). The improved temporal and spatial resolutions of the new generation sensors help researchers to understand in detail the kinematics of slope instability processes as well as the spatial and temporal patterns of landslide movement/activity and their relationships to causative or triggering factors [15].

The availability of data provided by different satellite sensors and the shortening of the revisiting time is a key to understanding the increasing success of the use of SAR images. Indeed, it was observed an increasing interest in the use of DInSAR for landslide detection and mapping in the last ten years [16].

Despite the widespread use of SAR with respect to the detection, mapping, modeling and monitoring of landslides, analysis dealing with the temporal variability of the landslide activity and rainfall trend has not been deepened. A few works dealing with the quantitative comparison between rainfall series and maps of surface displacements obtained from interferometric analyses of satellite data can be found in the scientific literature. Most of the studies are at the slope (or small catchment) scale and look for a comparison, mainly by means of graphs, between displacement series of a landslide and weekly, monthly or annual accumulated rainfall data gathered from a rain gauge more or less close to the landslide location, e.g., [17–20]. In some cases, other information from onsite measurements, as inclinometers, is included in the comparison with monthly and annual rainfall series, e.g., [21–23]; in other cases, the displacements maps are resampled to identify clusters with anomalous behavior, e.g., [24]. However, a few statistical correlations are searched or investigated. As an example, Ardizzone and co-authors [25], analyzing the Ivancich landslide, a deep-seated translational slide located in the municipality of Assisi (central Italy), evaluated the cross-correlation between the monthly rainfall data and the time series of the monthly ground displacement. They observed a complex temporal interaction between the rainfall and the ground deformation histories in the landslide area, with a lack of a direct effect of rainfall on the ground deformation.

A few examples are related to analyses conducted at the catchment, e.g., [26,27] in Spain, or the regional scale, e.g., [28] in Dominica, in which again only graphic comparisons between average displacements and weekly and monthly rainfall from one station are reported. Recently, Muñoz-Torrero Manchado and co-authors [29] searched for a correlation (calculated by means of the Pearson coefficient) between the annual number of shallow

landslides included in a regional inventory for the period 1992–2018 in western Nepal and seven climatic variables calculated using the ERA5 datasets. However, in this case, the landslide information is not retrieved from interferometric analysis but with the visual and systematic analysis of Google Earth images, also due to the type of investigated phenomena, i.e., shallow landslides.

To our knowledge, works dealing with the statistical comparison of ground deformations obtained from interferometric analysis of satellite-gathered data and rainfall measurements at the regional or catchment scale are not present in the scientific literature. With this paper, a procedure is proposed to fill this gap. Three datasets of Sentinel-1 images for the period 2015–2020 are processed by the Parallel Small Baseline Subset (P-SBAS) technique and analyzed with respect to mapped landslides. The obtained displacement series are analyzed and compared to the rainfall series to search for relationships by employing statistical and non-parametric tests, and to evaluate the effects of climate drivers on slow-moving landslides.

This methodology and its application were developed and carried out within the framework of the OT4CLIMA project (Development of Innovative Earth Observation Technologies for the Study of Climate Change and Its Impacts on the Environment), funded by the Italian Ministry of Education, University and Research, in the National Operational Program (PON), Research and Innovation 2014–2020, in the "Aerospace" thematic domain. The main aim of the project was to develop advanced Earth observation technologies and methodologies for improving the capabilities to better understand the effects of climate change and to mitigate them at the regional and sub-regional scale.

The procedure presented here was applied in the Basento river basin, within the Basilicata region, southern Italy. After an overall analysis of the whole basin, the defined procedure allows for the selection of two landslides to analyze their displacement series and search for correlations among displacements and rainfall series by means of statistical and non-parametric tests.

## 2. Study Area

### 2.1. Geological and Geomorphological Setting

From a geological point of view, the Basilicata region (Figure 1) is part of the southern Apennines chain. This chain is a NW–SE-oriented segment of the Italian Apennines and consists of an Adriatic-verging accretionary wedge derived from the Neogene compressional deformation of the Africa–Apulian passive margin, strongly dismembered by Quaternary neotectonics and therefore articulated in longitudinal and transversal basins [30,31]. The region is characterized by three different units from west to east: (1) the Apennines Chain with silico and carbonate deposits, evaporites and ophiolitic deposits (Upper Jurassic–Cretaceous to Middle Pliocene); (2) the Bradanic Trough, with Lower Pliocene and Quaternary deposits (gravel, clay and sand); and (3) Apulian Foreland characterized by well-layered Cretaceous limestones [32]. Due to its geological and tectonic features, the Basilicata region is one of the most prone Italian regions to geo-hydrological phenomena [33,34]. A recent regional-scale landslide inventory map shows a strong control of topographical and litho-technical features on the spatial distribution of landslides in Basilicata [35]. Moreover, the authors report that the percentage of the area affected by landslides is 7.7% of the total surface of the region (Figure 1) and that the most representative (i.e., about 55% of the total landslide area) landslides are earth flows (36.2%). Extreme rainfall or snowmelt occurrences are the main triggering factors of these phenomena [35–38]. Nearly 50% of the towns in the region are classified at high landslide or flood risk [39].

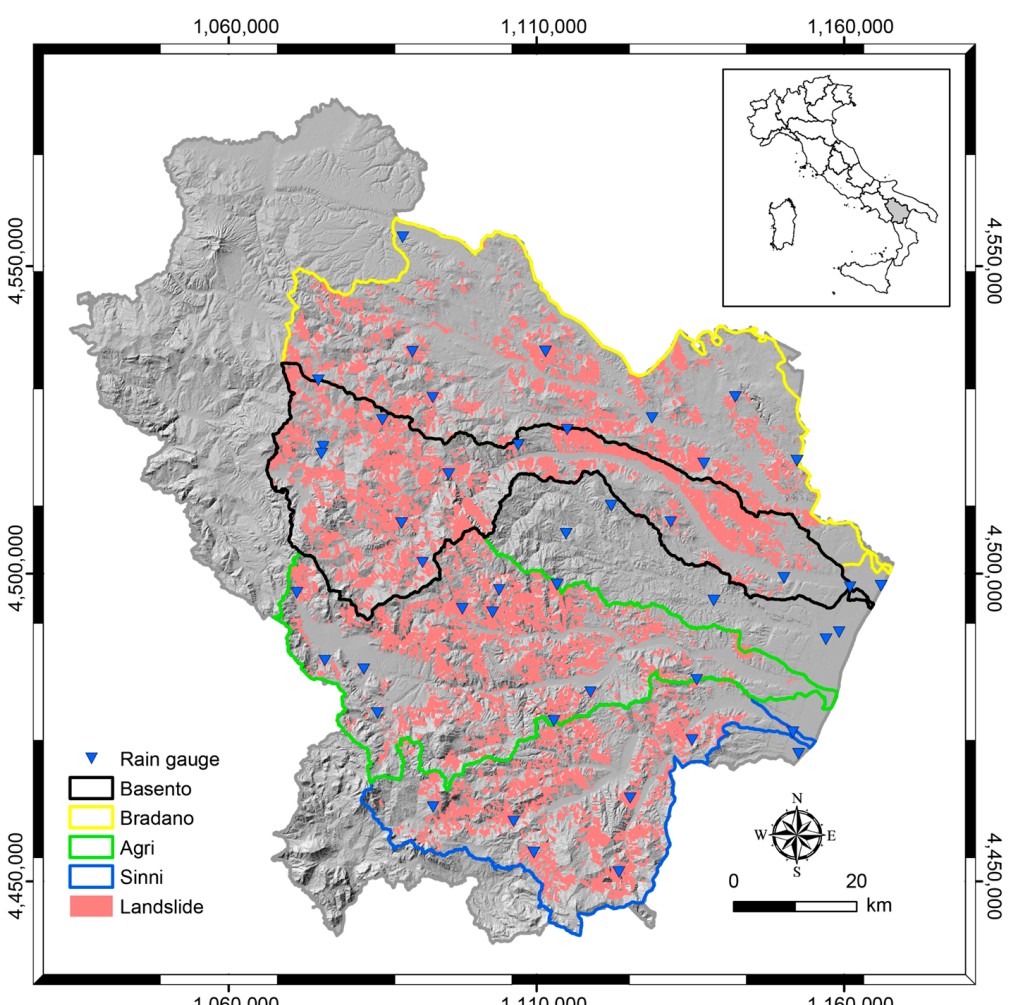

**Figure 1.** The inset shows the location of the Basilicata region. The main map shows the Basento, Bradano, Agri and Sinni River catchments. Landslides from the inventory of landslide phenomena in Italy (pink polygons) and rain gauges (blue triangles) used for rainfall analyses are also shown.

The Basento catchment, with an area of 1535 km$^2$ and a NW-SE trend, falls within the domain of the Apennine chain in the western part, and within the Bradanic Trough in the eastern part. The stratigraphic and structural setting of the basin plays a significant role in determining landslide occurrence and distribution. From the available bibliographic data, it appears that in the areas where highly overconsolidated Pliocene clays and intensely fissured tectonized Miocene clay shales outcrop, the most widespread landslides are slow earth flows, complex/compound landslides and, to a minor extent, rotational slides. Indeed, Guida and Iaccarino [40] recognized and mapped 484 landslides in the upper Basento catchment, mostly involving Pliocenic highly overconsolidated clays and Miocenic intensely fissured tectonized clay shales. According to Urciuoli et al. [41] about 95% of those landslides can be classified as earthflows, having a kinematic evolution characterized by mobilization and flow within a defined lateral shear surface, evolving in a very slow slide [23]. The particular geological context makes this area susceptible to reactivation phenomena.

*2.2. Climatic Features and Trends*

The Basilicata region has a typical Mediterranean temperate climate, with dry and warm summers (Csa and Csb, according to the Köppen–Geiger climate classification [42]). The mean annual rainfall is about 1000 mm and is mainly concentrated in autumn and spring [43]. A decreasing trend in annual and seasonal (mainly autumn–winter) total

precipitation was detected in the region in the period 1951–2010, particularly in 1970–2000, by using the Mann–Kendall non-parametric test (downward trend observed in 49 out of 55 analyzed stations) [43]. At the same time, an increase in the frequency and intensity of multi-day sequences of moderate to heavy precipitation, especially in the last decade, was observed. Such an increase has led to the growth of severe flooding and landslide events, not only in autumn and winter, but even in the early spring [43].

For a general indication of rainfall trend for the purposes of the OT4CLIMA project the Innovative Trend Analysis (ITA) method has been applied. The method, firstly proposed by Şen [44], unlike other methods (such as the well-known Mann–Kendall test), has the advantage that it does not require any assumptions (serial correlation, non-normality, sample number and so on). Moreover, its application is very simple [44–46] and it has been already applied in southern Italy [47]. Briefly, the time series is divided into two equal parts, with adequate length, which are separately sorted in ascending order; the two obtained sub-series are plotted on the X- and on Y-axis of a Cartesian plane, respectively, classified into three magnitude ranges (low, medium and high). If the data are collected on the 1:1 ideal line (45° line), there is no trend in the time series. If sample points are clustered in the upper triangular area of the 45° line, an increasing trend in the time series exists; conversely, there is a decreasing trend if data are in the lower triangular area. In this way, for any hydrometeorological or hydro-climatic time series, trends of low, medium and high values of data can be clearly identified.

Table 1 reports a summary of the ITA test for annual and seasonal rainfall totals and maxima measured by four stations in Basilicata, selected considering the minimum percentage of missing data in the period 1989–2018. Figure 2 shows, as an example, the results obtained for the series of maximum seasonal rainfall recorded by the Potenza station. Figure 2a reveals a positive trend for the whole data of the March–April–May (MAM) season. Figure 2b shows a weak positive trend for the low data and a negative trend for the medium and high part of the September–October–November (SON) data. The series of annual and seasonal rainfall totals (Table 1) show clear positive trends for almost all temporal aggregations (especially for the annual and winter series) and for almost all ranges of values. The same homogeneity is missing for what regards the series of maxima. However, the positive trend prevails, and is more evident for the MAM aggregation. It should be acknowledged that such analysis is derived from the comparison of two 15-year subseries, the first of which includes years characterized by relevant, widespread meteorological droughts.

**Table 1.** Trends, defined with the ITA method, for annual and seasonal rainfall totals and maxima for four stations in Basilicata (related basins in brackets) and for three ranges of values (low, medium and high). Key: DJF, December–January–February; MAM, March–April–May; JJA, June–July–August; SON, September–October–November; +, upward trend; -, downward trend; =, no trend.

| Station (Basin) | | Year | | DJF | | MAM | | JJA | | SON | |
|---|---|---|---|---|---|---|---|---|---|---|---|
| | | Total | Max | Total | Max | Total | Max | Total | Max | Total | Max |
| Matera (Bradano) | low | + | = | + | = | + | + | + | = | + | + |
| | med | + | = | + | = | + | + | + | = | + | = |
| | high | + | + | + | = | + | + | - | = | + | = |
| Potenza (Basento) | low | + | - | + | = | = | = | = | - | = | = |
| | med | + | + | + | = | + | + | + | + | + | - |
| | high | + | - | + | + | + | = | + | - | + | - |
| San Nicola (Basento) | low | + | + | + | + | + | + | + | + | + | + |
| | med | + | + | + | = | + | + | + | + | + | - |
| | high | + | + | + | + | + | + | + | + | + | = |
| Tramutola (Agri) | low | + | + | + | + | + | + | + | + | + | - |
| | med | + | + | + | + | + | + | + | + | = | - |
| | high | + | - | + | = | = | = | = | + | - | - |

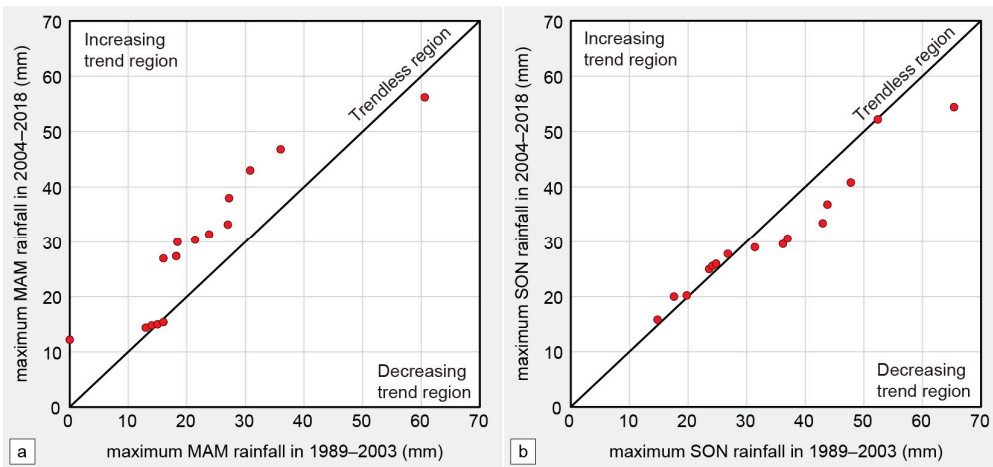

**Figure 2.** Results of the ITA test applied to the series of seasonal maxima recorded by the Potenza station, for the (**a**) March–April–May (MAM) and (**b**) September–October–November (SON) seasons.

## 3. Materials and Methods

### 3.1. DInSAR Analysis

Among the several multitemporal DInSAR techniques, in this work the Parallel Small Baseline Subset (P-SBAS) approach [48,49] was applied as implemented by Manunta et al. [50] to process Sentinel-1 IWS data. It is worth noting that with respect to [50] the processing chain has been slightly modified by introducing an additional step aimed at identifying and filtering out possible residual atmospheric artifacts that may affect the DInSAR measurements. This result is achieved through the proper exploitation of the available GNSS position time series. Moreover, from the algorithmic point of view, the automation and robustness of the Sentinel-1 P-SBAS processing chain was improved by introducing several check mechanisms to guarantee its fully unsupervised execution. The Sentinel-1 P-SBAS deformation time series were widely validated through comparisons with LOS-projected GNSS measurements [50,51]; these analyses demonstrated that the standard deviation of the difference between Sentinel-1 P-SBAS and GNSS measurements is <0.5 cm.

In the following, the main procedures characterizing the P-SBAS chain and the basic rationale of its implementation is described, without detailing the implemented algorithmic solutions and the parallelization techniques exploited for each processing step, which are extensively discussed in [50–52]. Figure 3 shows a schematic flowchart of the P-SBAS processing chain.

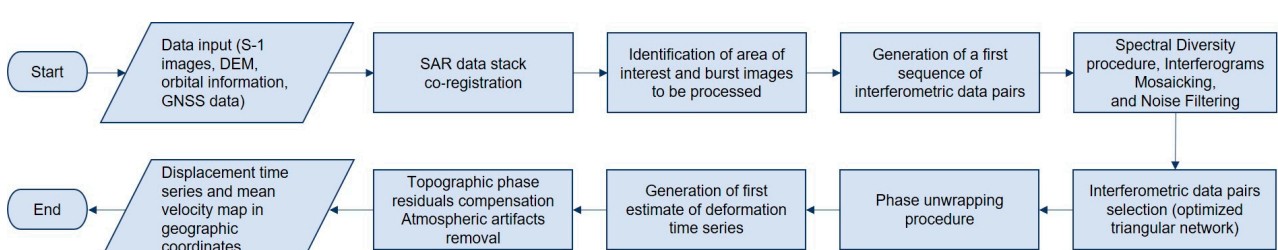

**Figure 3.** Flowchart of the P-SBAS processing chain. Modified from [50,51].

The first operations concern handling and ingestion of input data, represented by the sequence of Sentinel-1 Single Look Complex (SLC) images, the orbital information associated to each SAR acquisition, the digital elevation model (DEM) of the investigated zone and the GNSS position time series available for the area, which in our case are provided by the Nevada Geodetic Laboratory at the University of Nevada, USA (UNR-NGL) [53].

Once the data input is correctly ingested, the SAR data stack is properly co-registered with respect to a reference SAR geometry by exploiting the geometrical SAR registration

procedure described in [50]. In particular, the co-registration procedure is carried out through the cascade of three steps, referred to as rigid offset estimation, master image rigid offset retrieval, and geometric registration, respectively. The first two steps of this procedure (i.e., the rigid offset estimation and master image rigid offset retrieval) allow us to precisely estimate the rigid shift between each SAR image and a selected reference one, referred to as the master image. The achieved results are subsequently used to correctly carry out the geometric registration step based on the approach presented by Sansosti et al. [54] that allows us to achieve a temporal sequence of co-registered burst images, by exploiting topographic and orbital information. Subsequently to the geometrical co-registration step, the area of interest as well as the burst images to be processed are identified and a first sequence of interferometric data pairs involving the available SAR acquisitions is created, which is then exploited within the subsequent interferograms generation, Spectral Diversity procedure and noise filtering operations [50]. At this stage, the corrected multi-look interferograms (and the corresponding spatial coherence maps) of adjacent bursts are accurately assembled through an interferogram mosaicking operation to generate the differential interferograms of the whole investigated area, which is hereafter referred to as the "frame". Moreover, on these interferograms, the noise filtering procedure discussed in [55] is applied.

Subsequently, a selection, among all the considered small baseline interferometric pairs, of an optimized triangular network within the perpendicular baseline/time plane is computed [55]. In particular, maximum values of 150 days for the temporal baseline and of 200 m for the spatial baseline were selected. This implies that essentially no constraint is applied to the spatial baselines because the orbital tube diameter is about 200 m. The sequence of the so-identified multi-look noise-filtered small baseline interferograms of the frame is then unwrapped through the extended minimum cost flow (EMCF) phase unwrapping (PhU) algorithm [56]. Once EMCF is carried and the stack of unwrapped interferograms is retrieved, a first estimate of the deformation time series is computed by applying the SVD method, following the lines of the original SBAS approach [48]. In this step, a procedure for the compensation of possible topographic phase residuals is implemented. Moreover, it includes the estimation and removal of atmospheric artifacts by taking into account that they are typically correlated in space and poorly in time [48], as well as by benefiting from their correlation with topography.

Last operations are represented by an additional step implemented within the presented P-SBAS processing chain, which is discussed in detail in [51]. In this block, the available GNSS measurements are used to identify and filter out possible residual atmospheric artifacts that may affect the DInSAR measurements [51]. Lastly, the final displacement time series and the corresponding mean deformation velocity maps of the investigated frame are computed and generated in a geographic/cartographic reference system.

In this work, three Sentinel-1 datasets, acquired by descending and ascending orbits over Southern Italy, were processed with the P-SBAS processing chain. In particular, data collected by Sentinel-1 Track 146 (ascending) and Tracks 51 and 124 (descending) between 2015 and 2020 were analyzed (Figure 4). No spatial baseline constraint was imposed in the interferometric pair selection exploited in our processing thanks to the narrow orbital tube characterizing the Sentinel-1 constellation. Moreover, the 1-arcsec SRTM DEM was exploited to generate the analyzed DInSAR interferograms on which a complex multi-look operation with 20 looks in range directions and 5 in the azimuth one were performed to obtain a pixel dimension of about $80 \times 80$ m. The same DEM was also used to geocode the computed deformation time series and the corresponding mean velocity maps. The achieved deformation time series and velocity maps provide information on the coherent pixels, identified by considering those with a temporal coherence value [56] greater than a selected threshold that in our analysis is set equal to 0.9 for all the processed Sentinel-1 datasets.

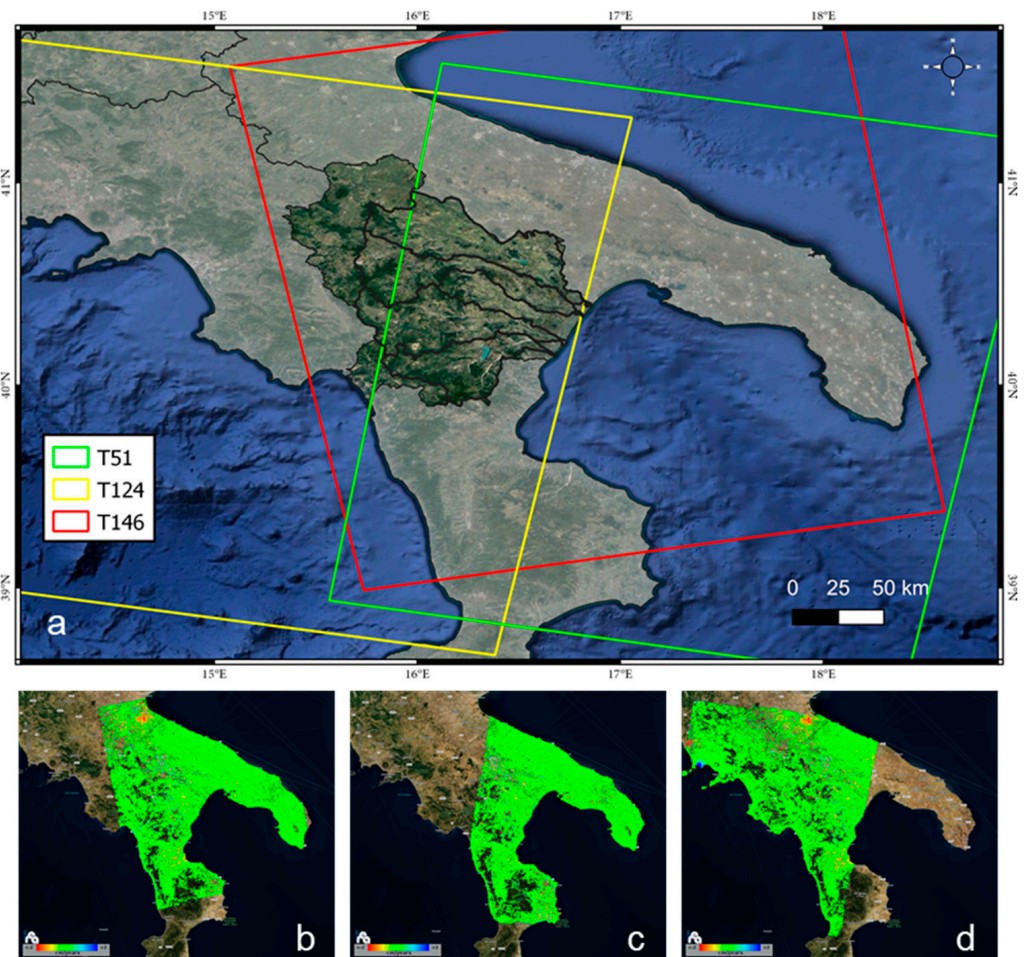

**Figure 4.** (**a**) Sentinel-1 tracks used in this work. Results of the P-SBAS analysis for the (**b**) T146, ascending, and (**c**) T51 and (**d**) T124, descending orbits. Background from Google Earth.

### 3.2. Geomorphological Analysis

The selection of the case studies (slow-moving landslides) from the achieved deformation time series and velocity maps, derived from the processing of Sentinel-1 images in ascending and descending orbits, was based on the identification of clusters of pixels (represented by points in the figures of the present work) that are characterized by similar behavior, from a cinematic point of view. The adopted selection procedure was based on geomorphological criteria and available data and information.

The available data that were used for the selection procedure are as follows:

- Inventory of landslide phenomena in Italy (IFFI), available in polygonal shapefile format (https://www.progettoiffi.isprambiente.it/ (accessed on 1 September 2022) and https://idrogeo.isprambiente.it/app/ (accessed on 1 September 2022)).
- Geological map of Italy, available as a WMS service on the Italian National Geoportal (http://wms.pcn.minambiente.it/ogc?map=/ms_ogc/WMS_v1.3/Vettor-ali/Carta_geolitologica.map; accessed on 1 September 2022).
- Maps of ground deformation and associated "time series" prepared for the OT4CLIMA project from Sentinel-1 images in ascending and descending orbits, for the time period 2015–2020, in shapefile format (point geometry).
- Map of the rain gauges and the daily rainfall measurements (provided by the Decentralized Functional Center of the Basilicata Region civil protection, http://www.centrofunzionalebasilicata.it/it/; accessed on 1 September 2022).

- Literature and technical documentation and consultation of local online newspapers related to landslide activity in the Basilicata region.

The selection of the case studies was carried out according to the following work phases:

1. In the GIS environment, the velocity maps were overlaid on the IFFI landslide shape-file, the geological map and the map of the rain gauges by keeping as background the Google Earth and/or Bing satellite image.

2. Through a GIS intersection, the pixels on the velocity maps that fell within a circular buffer of a 10 km radius from the rain gauges were selected. This radius can be considered as a probable influence range of the rain on landslides, it is also in accordance with several works dealing with the reconstruction of rain-gauge-based rainfall events able to trigger landslides in Italy (e.g., [57] and references therein)

3. The selected pixels were classified based on the average velocity of deformation (Figure 5a) over the five years of observation (2015–2020) and the cumulative measure of deformation at the last measurement date (Figure 5b). In the average velocity classification, the pixels characterized by a velocity between −0.1 and 0.1 cm/year were considered "stationary" and excluded from the further analyses.

4. The pixels with analogous increasing or decreasing trends located in or around landslide areas were put in clusters.

5. Information on landslide activities from the scientific literature, online newspapers and technical documents were analyzed to select the landslides characterized by the same state of activity.

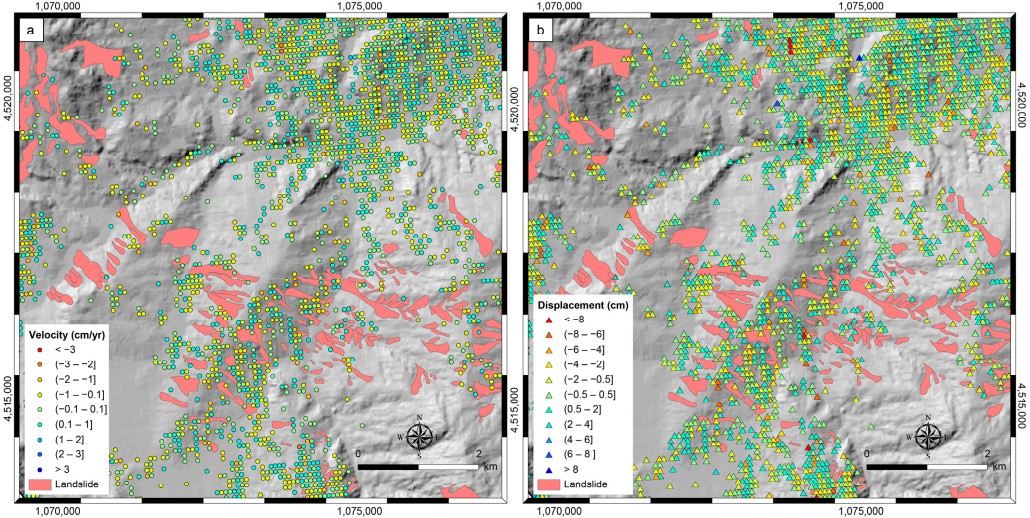

**Figure 5.** Classifications for (**a**) average velocity of deformation over the five years of observation (2015–2020) and (**b**) cumulative deformation at the last measurement date, adopted in this work.

### 3.3. Statistical Analysis

Global and time lagged synchronies between the two time series data were investigated and quantified, as outcome of interactions between rainfall (the forcing/cause) and surface displacement (effect). Indices that do not make assumptions on the probability density functions of the series were adopted, because the rainfall data and their logarithm are not even roughly normally distributed, and they show an uneven variability (see similar concepts of homoscedasticity).

Thus, the Kendall rank correlation coefficient $\tau$ [58], and the information based Maximal Information Coefficient (MIC) [59] were adopted in the analysis. The classical Pearson's correlation coefficient was not adopted because the variables here analyzed, in particular the rainfall, were not normally distributed, as requested by the method. On the other hand, the Kendall rank correlation coefficient does not need strong assumptions on the variable distributions.

The first, $\tau$, is a statistic measuring the relationship, or the association between rankings of different ordinal variables (e.g., rainfall and displacement), and it can be used to measure the significance of their relation.

It is obtained through the following equation:

$$\tau = \frac{P - Q}{\sqrt{(P + Q + T)(P + Q + U)}} \tag{1}$$

where $P$ is the number of concordant pairs, $Q$ the number of discordant pairs, $T$ the number of ties in $\bar{x}$ (our first data series ranking) and $U$ in $\bar{y}$ (our second data series ranking). A pair of observations $(x_i, y_i)$ and $(x_2, y_2)$ where $i < j$ is concordant when $x_i > x_j$ and $y_i > y_j$, or $x_i < x_j$ and $y_i < y_j$. The pair is discordant when $x_i > x_j$ and $y_i < y_j$, or $x_i < x_j$ and $y_i > y_j$ [60]. An increasing $\tau$ implies increasing agreement between the two series rankings. The two series can be considered as statistically associated when the null hypothesis ($H_0$) of independence is rejected. In this work, a *p*-value lower than 0.1 ($p < 0.1$) was considered as strong evidence against the independence to get less than 10% of probabilities that $H_0$ is correct. In particular, $\tau$ was computed through the 1.8.0 version of the *scipy.stats.kendalltau* function (to check whether it is a function or not) in Python v. 3.7.

The latter, the MIC, is an exploratory data analysis tool that looks for the scatter plot binning of a pair of variables (e.g., rainfall and displacement) maximizing their induced mutual information [61] where the information is measured through the number of points falling inside the boxes of the scatterplot [59]. MIC is obtained through the following equation:

$$MIC = max\left\{I(x,y)/log_2 min\{n_x, n_y\}\right\} \tag{2}$$

where

$$I(x,y) = \sum_{i=1}^{n_x} p(x_i) log_2 \frac{1}{p(x_i)} + \sum_{j=1}^{n_y} p(y_j) log_2 \frac{1}{p(y_j)} - \sum_{i=1}^{n_x} \sum_{j=1}^{n_y} p(x_i y_j) log_2 \frac{1}{p(x_i y_j)} \tag{3}$$

$\bar{x}$ and $\bar{y}$ are the series and $n_x$ and $n_y$ are the number of the bins of the partitions of the x- and y-axis, respectively [62]. MIC does not rely on distributional assumptions and it is able to identify a broad class of associations [62] with scores ranging from 0 (no association) to 1 (full association). The MIC was estimated using the 1.5.10 version of the *cstats* function in the 'Minerva' package, in R v. 3.6.3.

## 4. Results

The application of the procedure described in Section 3 allowed the selection of clusters of pixels, in or near landslide areas, and the identification of significant case studies among these clusters.

After the overlay of the IFFI landslide shapefile and the lithological map with the DInSAR-based displacement pixels, the rain gauges of Potenza QA, Laurenzana and Albano di Lucania, within the Basento river catchment, were selected. Then, the ground displacement pixels (based on the ascending and descending satellite data) within a circular buffer of 10 km in radius centered in each of the three rain gauges, were selected, allowing the identification of 13 clusters of pixels located in or near landslide areas (taking into account that the pixels could partially overlap the landslide polygons). Among these clusters, two landslide areas characterized by the same state of activity were selected as significant case studies, also exploiting information gathered from the scientific literature, newspapers and technical documents.

The first landslide (Landslide 1 in Figure 6) is a rotational slide with a maximum width of 0.4 km and length of about 1.2 km, while the second (Landslide 2 in Figure 6) is a slow earth flow with a maximum width of 0.4 km and length of about 1.4 km. The landslide areas involve agricultural land, mainly arable, with dispersed housing and roads. More in detail, according to the 2018 CORINE Land Cover map, Landslide 1

involves only non-irrigated arable land (code 211), while Landslide 2 involves mostly complex cultivation patterns (code 242) (Figure 6b). Moreover, according to the Italian geo-lithological map, both landslides occur within clayey and clayey-limestone (turbidites) units of the Paleocene (Figure 6c). Landslide 1 occurs for a minor part in in sands and conglomerates of the Pliocene.

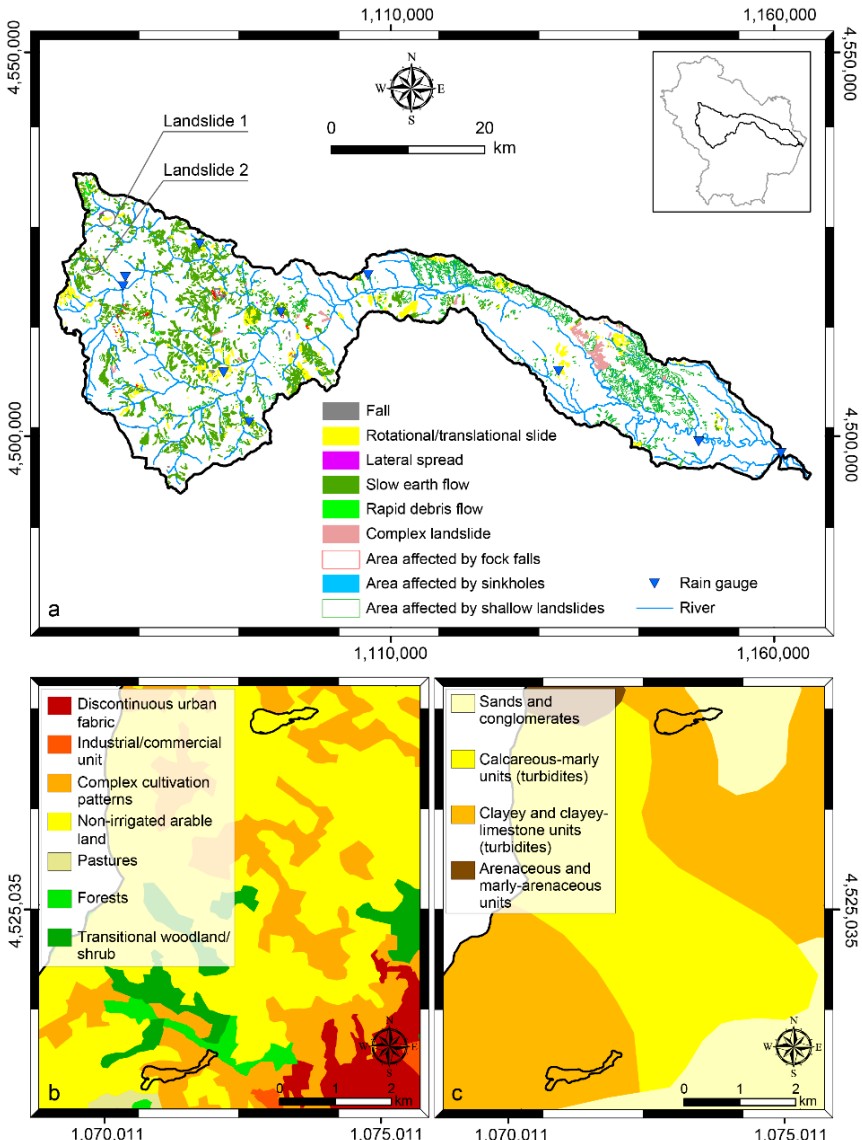

**Figure 6.** (**a**) Boundaries of the Basento river catchment with indications of the main landslide types according to the IFFI database (https://idrogeo.isprambiente.it/app/; accessed on 1 September 2022), of the rain gauges and of the two selected landslides. (**b**) Land cover of the areas of the selected landslides according to the 2018 CORINE Land Cover map (https://land.copernicus.eu/pan-european/corine-land-cover/clc2018; accessed on 1 September 2022). (**c**) Geo-lithological settings of the areas of the selected landslides according to the Italian geo-lithological map (http://www.pcn.minambiente.it/mattm/servizio-di-scaricamento-wfs/; accessed on 1 September 2022).

With reference to the boundaries of the selected rotational slide and earth flow, Figures 7 and 8 show the surface displacements for the T146 ascending orbit and T124 descending orbit, respectively, classified according to the average velocity of deformation over observation period.

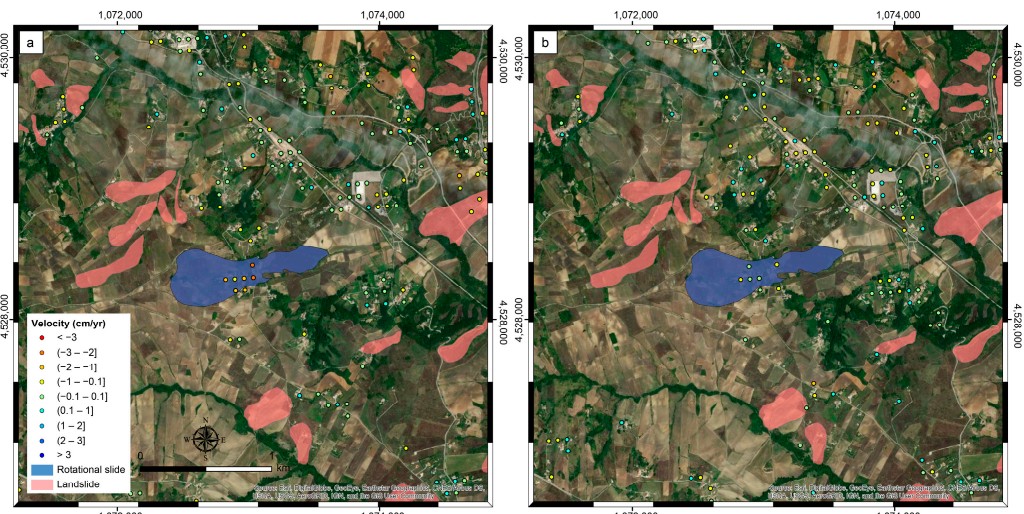

**Figure 7.** Boundary of the rotational slide (landslide 1 in Figure 6), with the points representing the pixels of surface displacements for the (**a**) T146 ascending orbit (9 pixels) and (**b**) T124 descending orbit (10 pixels), classified according to the average velocity of deformation over the five years of observation (2015–2020). Background from Google Earth.

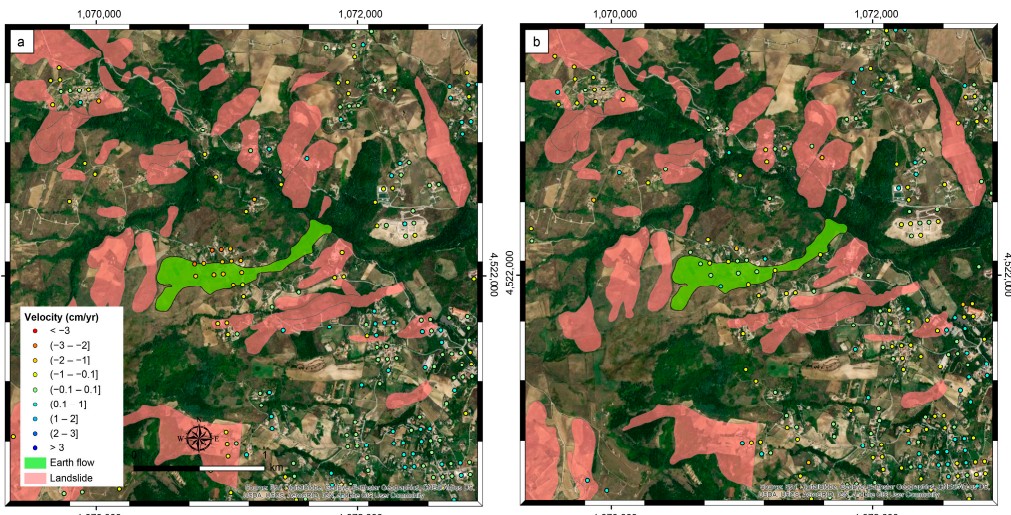

**Figure 8.** Boundary of the earth flow (landslide 2 in Figure 6), with the points representing the surface displacements for the (**a**) T146 ascending orbit (11 pixels) and (**b**) T124 descending orbit (10 pixels), classified according to the average velocity of deformation over the five years of observation (2015–2020). Background from Google Earth.

Figure 9 shows an example of the results related to the selected rotational slide (Figure 7), for the T124 descending orbit. Among all the series of cumulative displacements for all the pixels falling within the landslide polygon (Figure 9a), an average, representative path was identified (considering only the pixels with analogous paths) and used for comparison with the series of rainfall accumulated over different periods. The displacement series are roughly normally distributed, therefore considering the average or the median values for defining the representative path is equivalent, because the differences between average and median values are not significant. The visual comparison between the representative cumulative displacements of the landslide and the 3-day cumulative rainfall (shown in Figure 9b as example) shows increases in the landslide displacements in correspondence of rainfall peaks. Moreover, the relative displacements and the series of daily rainfall accumulated over different periods were plotted on a Cartesian plane (Figure 9c shows the relative displacements vs. the 3-day cumulative rainfall) to search for

visual and statistical relationships. Finally, the Kendall test was carried out considering the relative displacements with different shifts (lag from 0 to 10 days between the rainfall and the day the displacement is measured from the satellite) and the cumulative rainfall series. Only the cumulative rainfall series that satisfied the hypothesis of a *p*-value lower than 0.1 are shown: Figure 9d shows that the series of 3-day cumulative rainfall has the highest score, with a shift of 5 to 7 days. This means that the higher correlation for this landslide activity can be observed for 3-day cumulative rainfall with a lag time of 5 to 7 days.

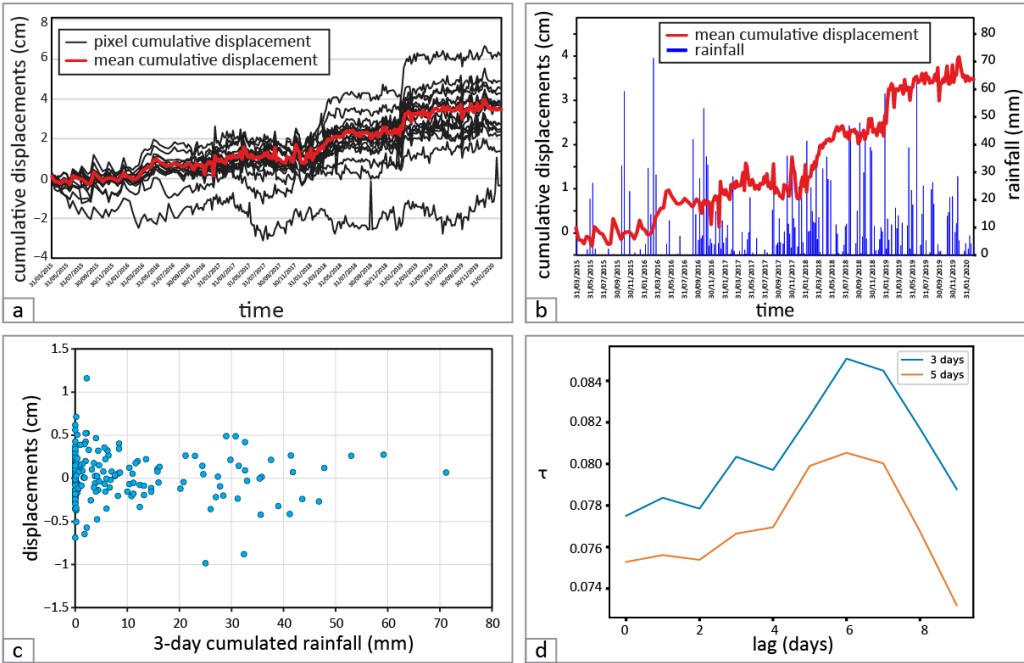

**Figure 9.** Example of the results obtained for the rotational slide (Figure 7), for the T124 descending orbit: (**a**) cumulative displacements measured over the whole observation period (black curves) with the indication of the average displacements representative for the landslide (red curve); (**b**) representative cumulative displacements and 3-day cumulative rainfall; (**c**) scatter plot of the relative displacements and the 3-day cumulative rainfall; (**d**) values of the Kendall coefficient, $\tau$, measuring the ordinal association between rainfall series and relative displacements with different shifts (lag from 0 to 10 days); only the rainfall series that satisfied the hypothesis of a $p < 0.1$ are shown (in this case only the series of 3-day and 5-day cumulative rainfall).

Analogously, Figure 10 shows an example of the results related to the selected earth flow (Figure 8), still for the T124 descending orbit. In this case, the best value of the Kendall coefficient pertains to the 7-day cumulative rainfall, regardless of the lag time with the landslide displacements. For the sake of brevity, only these two results are reported here; however, the method can be applied to every orbit and every rainfall series.

Figure 11 shows the values of the MIC for different cumulative rainfall (3 days to 30 days) and different shifts of the displacements–rainfall series (lag from 0 to 10 days), for the two considered landslides and the two interested orbits (T124, desc and T146, asc). Greenish cells indicate a higher MIC, meaning a higher exchange of information between the two series. Remarkably, the highest values of the MIC (cells in green) are related to a shift from 5 to 8 days and a 20- to 30-day cumulative rainfall. It is the case of the rotational slide and the T146 ascending orbit (Figure 11b). However, the values of the MIC related to a 3- to 5-day cumulative rainfall and a shift from 5 to 7 days (the combination that obtained the highest *p*-value) are still high. Moreover, it can be observed that there is a higher rate of shared information in the series related to the rotational landslide (Figure 11a,b) than those related to the translational landslide (Figure 11c,d).

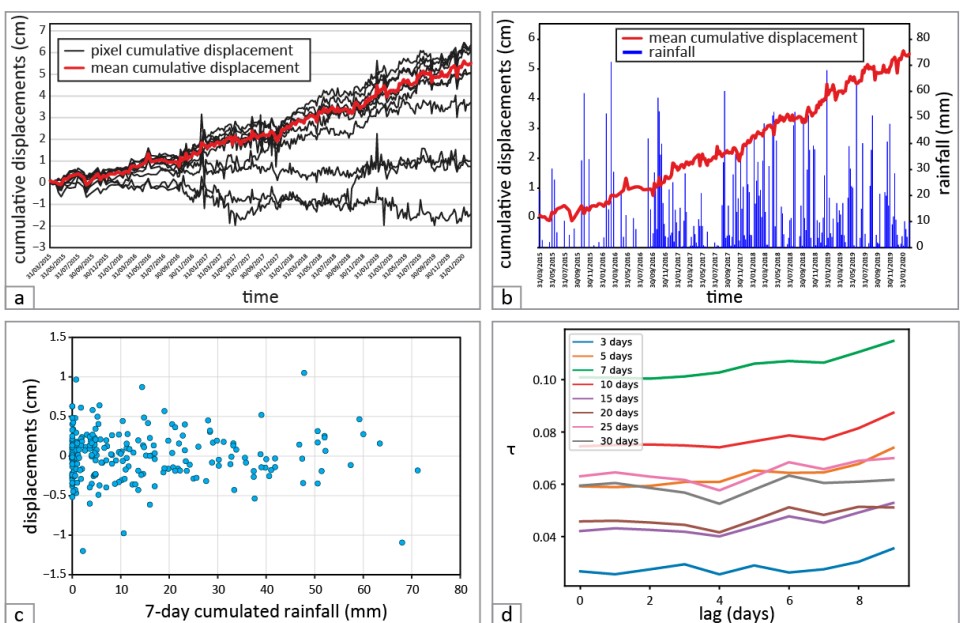

**Figure 10.** Example of the results obtained for the earth flow (Figure 8), for the T124 descending orbit: (**a**) cumulative displacements measured over the whole observation period (black curves) with the indication of the average displacements representative for the landslide (red curve) defined excluding the three lower paths that have a trend different from the others; (**b**) representative cumulative displacements and 7-day cumulative rainfall; (**c**) scatter plot of the relative displacements and the 7-day cumulative rainfall; (**d**) values of the Kendall coefficient, $\tau$, measuring the ordinal association between rainfall series and relative displacements with different shifts (lag from 0 to 10 days); only the rainfall series that satisfied the hypothesis of $p < 0.1$ are shown.

**(a)**

|  | 3 d | 5 d | 7 d | 10 d | 15 d | 20 d | 25 d | 30 d |
|---|---|---|---|---|---|---|---|---|
| 0 | 0.191 | 0.203 | 0.170 | 0.161 | 0.167 | 0.178 | 0.172 | 0.176 |
| 1 | 0.188 | 0.163 | 0.181 | 0.193 | 0.193 | 0.185 | 0.172 | 0.196 |
| 2 | 0.197 | 0.191 | 0.191 | 0.193 | 0.175 | 0.180 | 0.178 | 0.181 |
| 3 | 0.192 | 0.193 | 0.181 | 0.185 | 0.193 | 0.200 | 0.186 | 0.194 |
| 4 | 0.172 | 0.177 | 0.177 | 0.171 | 0.172 | 0.167 | 0.202 | 0.189 |
| 5 | 0.171 | 0.185 | 0.220 | 0.221 | 0.230 | 0.218 | 0.209 | 0.186 |
| 6 | 0.172 | 0.202 | 0.198 | 0.174 | 0.221 | 0.172 | 0.157 | 0.167 |
| 7 | 0.175 | 0.186 | 0.185 | 0.194 | 0.191 | 0.204 | 0.202 | 0.186 |
| 8 | 0.226 | 0.178 | 0.200 | 0.241 | 0.192 | 0.222 | 0.202 | 0.191 |
| 9 | 0.183 | 0.213 | 0.230 | 0.212 | 0.228 | 0.255 | 0.273 | 0.239 |
| 10 | 0.199 | 0.164 | 0.214 | 0.185 | 0.192 | 0.195 | 0.220 | 0.246 |

**(b)**

|  | 3 d | 5 d | 7 d | 10 d | 15 d | 20 d | 25 d | 30 d |
|---|---|---|---|---|---|---|---|---|
| 0 | 0.195 | 0.188 | 0.211 | 0.232 | 0.258 | 0.271 | 0.274 | 0.289 |
| 1 | 0.186 | 0.206 | 0.243 | 0.219 | 0.236 | 0.284 | 0.254 | 0.277 |
| 2 | 0.210 | 0.227 | 0.250 | 0.244 | 0.292 | 0.254 | 0.290 | 0.284 |
| 3 | 0.185 | 0.197 | 0.234 | 0.196 | 0.226 | 0.244 | 0.262 | 0.237 |
| 4 | 0.169 | 0.185 | 0.198 | 0.213 | 0.219 | 0.287 | 0.340 | 0.326 |
| 5 | 0.163 | 0.190 | 0.213 | 0.259 | 0.260 | 0.275 | 0.318 | 0.340 |
| 6 | 0.213 | 0.207 | 0.209 | 0.240 | 0.262 | 0.297 | 0.373 | 0.408 |
| 7 | 0.183 | 0.222 | 0.225 | 0.237 | 0.264 | 0.315 | 0.368 | 0.424 |
| 8 | 0.226 | 0.302 | 0.318 | 0.308 | 0.270 | 0.290 | 0.326 | 0.327 |
| 9 | 0.177 | 0.212 | 0.229 | 0.231 | 0.230 | 0.271 | 0.282 | 0.313 |
| 10 | 0.226 | 0.242 | 0.251 | 0.286 | 0.224 | 0.253 | 0.298 | 0.330 |

**(c)**

|  | 3 d | 5 d | 7 d | 10 d | 15 d | 20 d | 25 d | 30 d |
|---|---|---|---|---|---|---|---|---|
| 0 | 0.212 | 0.180 | 0.206 | 0.220 | 0.219 | 0.186 | 0.197 | 0.200 |
| 1 | 0.156 | 0.171 | 0.174 | 0.198 | 0.180 | 0.213 | 0.202 | 0.206 |
| 2 | 0.194 | 0.180 | 0.177 | 0.194 | 0.205 | 0.225 | 0.232 | 0.222 |
| 3 | 0.216 | 0.195 | 0.208 | 0.197 | 0.166 | 0.155 | 0.201 | 0.164 |
| 4 | 0.195 | 0.198 | 0.175 | 0.178 | 0.182 | 0.188 | 0.218 | 0.194 |
| 5 | 0.179 | 0.189 | 0.170 | 0.195 | 0.202 | 0.184 | 0.229 | 0.214 |
| 6 | 0.173 | 0.199 | 0.189 | 0.179 | 0.215 | 0.182 | 0.191 | 0.176 |
| 7 | 0.165 | 0.190 | 0.218 | 0.170 | 0.188 | 0.209 | 0.174 | 0.181 |
| 8 | 0.213 | 0.245 | 0.201 | 0.241 | 0.223 | 0.211 | 0.266 | 0.271 |
| 9 | 0.202 | 0.201 | 0.188 | 0.157 | 0.188 | 0.209 | 0.215 | 0.226 |
| 10 | 0.197 | 0.176 | 0.188 | 0.189 | 0.189 | 0.229 | 0.191 | 0.209 |

**(d)**

|  | 3 d | 5 d | 7 d | 10 d | 15 d | 20 d | 25 d | 30 d |
|---|---|---|---|---|---|---|---|---|
| 0 | 0.175 | 0.188 | 0.192 | 0.187 | 0.164 | 0.181 | 0.169 | 0.168 |
| 1 | 0.192 | 0.193 | 0.215 | 0.215 | 0.197 | 0.169 | 0.167 | 0.151 |
| 2 | 0.203 | 0.170 | 0.169 | 0.161 | 0.172 | 0.188 | 0.188 | 0.171 |
| 3 | 0.151 | 0.164 | 0.172 | 0.188 | 0.176 | 0.167 | 0.186 | 0.167 |
| 4 | 0.204 | 0.205 | 0.165 | 0.204 | 0.182 | 0.180 | 0.167 | 0.164 |
| 5 | 0.206 | 0.185 | 0.182 | 0.156 | 0.152 | 0.188 | 0.166 | 0.156 |
| 6 | 0.187 | 0.175 | 0.193 | 0.179 | 0.211 | 0.212 | 0.182 | 0.220 |
| 7 | 0.229 | 0.187 | 0.190 | 0.197 | 0.196 | 0.232 | 0.202 | 0.221 |
| 8 | 0.239 | 0.220 | 0.182 | 0.197 | 0.166 | 0.185 | 0.220 | 0.203 |
| 9 | 0.186 | 0.224 | 0.200 | 0.233 | 0.224 | 0.244 | 0.204 | 0.214 |
| 10 | 0.197 | 0.241 | 0.229 | 0.187 | 0.211 | 0.197 | 0.181 | 0.200 |

min           max

exchange of information

**Figure 11.** Values of the MIC for different cumulative rainfall (3 days to 30 days) and different shifts of the displacements–rainfall series (lag from 0 to 10 days). Results for the rotational slide (Figure 7) related to the (**a**) T124 descending orbit and (**b**) T146 ascending orbit, and for the earth flow (Figure 8) for the (**c**) T124 descending orbit and (**d**) T146 ascending orbit. Greenish cells indicate a higher MIC (higher exchange of information between the two series) compared to the reddish ones (lower MIC).

## 5. Discussion

The procedure proposed in this work allows for the identification of clusters of pixels measuring satellite-based surface displacements indicating landslide activity, and for the comparison among series of displacements and rainfall measurements.

In the work, we faced different problems related to series completeness and spatial/temporal data density. On one hand, rainfall series contained some gaps that did not allow the proper application of well-known tests, such as the non-parametric Mann–Kendall test, for the identification of trends. At the same time, the ITA test was applied to two relatively recent periods and only to the series that had sufficient data. However, the rainfall series fully overlap temporally the satellite-based displacement measurements, thus allowing a comparison among them.

On the other hand, the availability of a landslide inventory map in digital format provided a starting point for the geomorphological analysis. The regional coverage of the inventory required a non-straightforward multi-scale analysis to identify the clusters of pixels on landslide areas. The portion of landslides that can be detected through DinSAR techniques in a territory can depend on several factors including type, exposition and/or displacement direction, velocity and land cover. However, it is generally low. Recently, Festa et al. [63] processed Sentinel-1 images acquired from March 2015 to December 2018 over the whole Italian peninsula through the P-SBAS technique. They compared the results with the landslides mapped by the IFFI project and observed that the fraction of deforming areas positioned on previously mapped landslides is 12.0% and 13.8% for descending and ascending orbit, respectively. Furthermore, ground displacement pixels obtained through InSAR analyses are mostly located along road infrastructure and at buildings in urban areas, where landslides are often absent because they are not reported in the inventory maps or, if present, already stabilized by anthropic intervention. Moreover, only for very few landslides was it possible to confirm the state of activity in the considered time period (2015–2020), therefore the number of landslides to be considered was greatly reduced. Additionally, the clusters of pixels that characterized the movement of a landslide had to have a similar trend and to be distinguishable from other pixels outside the landslide itself, which do not show any remarkable displacement. Finally, it was necessary to be sure that the landslide activity in the analyzed period was related to rainfall, and for this reason the buffer around rain gauges was considered. Indeed, Basilicata is a region where landslide triggers also include other factors such as snowmelt [64] and earthquake [65–67]. Therefore, in order to consider only rainfall-related landslides and avoid misinterpretations, the landslide selection must be carried out by gathering as much information as possible. Regarding the snowmelt, to our knowledge no other activities related to snowmelt were found for the two considered landslides in the observation period. Regarding the seismic trigger, no landslides triggered by earthquakes in Basilicata in the period between 2015–2022 were listed in the Italian Catalog of Earthquake-induced ground failures [67].

The use of Sentinel-1 images allowed the analysis of a 5.5-year time window (2015–2020). Moreover, the Sentinel-1 constellation appears to be a suitable choice because it offers an almost global coverage, and then the possibility of reproducing this method elsewhere, and an unprecedented revisit time, i.e., a 6-day repeat cycle at the latitude of the study area. This high revisit frequency allows obtaining a series of close measurements of displacements able to evaluate the activity of slow-moving landslide phenomena. An analysis aimed at the evaluation of long-lasting variations would need a time series longer than the one adopted here. The use of other satellite-based products, such as ERS or Envisat, would have allowed the analysis of a longer time series, with the hamper of a lower revisit frequency (around 30 days) and a covering period ending in 2012. The construction of a merged series with different products is not trivial and, in any case, would lead to a reduction in the temporal resolution of the series. For these reasons, we preferred to test our procedure with this relatively short temporal period, with the idea of extending it as soon as the Sentinel-1 program reaches a longer covering period. In fact, it is expected that a longer observation

period and an increase of acquisition frequency (e.g., daily) of the images would allow a better comparison between rainfall and displacement measurements and then a deeper understanding of the underlying mutual relationship.

It is worth noting that, due to several phenomena (mainly geometrical distortions and temporal decorrelation effects), steep and vegetated areas could not be observable with InSAR techniques. In this work, we have not used a visibility index [68], usually adopted to investigate areas in which InSAR can be successfully employed to analyze slope processes, including landslides. The use of visibility indices is suggested when there is scarcity of data and/or knowledge of the study area. In our case, having already information on both landslides and displacements, we did not need to verify *a priori* whether DInSAR was useful or not.

## 6. Conclusions

The main aim of this work was to define and propose for the first time in the literature a quantitative procedure for identifying satellite-based displacement clusters, comparing them with rainfall series (from different sources) and applying statistical tests to evaluate their relationships at a regional scale. In particular, for the analyzed test case, the results show that, for both landslides, there is a significance of the Kendall's test for the displacements detected by the descending orbit (T124) for a 3-day cumulative rainfall and a slope response of about 7 days. Although the study area was chosen in the framework of the OT4CLIMA project, and thus, not specifically identified for landslide analyses, it was possible to get to the point of selecting two case studies (the landslides) in order to apply a statistical analysis. The proposed statistical tests here used to evaluate the relations between weather forcing and displacements might be also applied to select optimal movement predictors necessary to develop data-driven models, e.g., [69,70], for simulating and eventually predicting displacement time-series.

Given the replicability of the whole quantitative procedure, we expect that it can be applied in any study area. Interesting future applications might include satellite- or radar-based rainfall estimates and might consider areas with different climatic and physiographic features and eventually also the analysis of other variables, such as air or rock temperature.

**Author Contributions:** Conceptualization, F.A., S.L.G., E.V., L.A., R.C., M.M. and A.C.M.; methodology, F.A., S.L.G., E.V., L.A., R.C., M.M. and A.C.M.; formal analysis, A.C.M., S.L.G., R.C. and M.M.; investigation, F.A., L.A. and E.V.; data curation, M.M., E.V., S.L.G. and F.A.; writing—original draft preparation, F.A., S.L.G., E.V., L.A., R.C., M.M. and A.C.M.; writing—review and editing, F.A., S.L.G., E.V., L.A., R.C., M.M. and A.C.M. All authors have read and agreed to the published version of the manuscript.

**Funding:** This research was funded, in the framework of the national project OT4CLIMA, by the Italian Ministry of Education, University and Research (MIUR), in the framework of the National Operational Programme (PON), Research and Innovation 2014–2020, Specialization Area Aerospace (D.D. 2261 del 6.9.2018, PON R&I 2014–2020 and FSC). Grant number: ARS01-00405.

**Data Availability Statement:** Not applicable.

**Acknowledgments:** The Multi-Risk regional center of Basilicata provided rainfall data. Authors thank Maurizio Lazzari (CNR-ISPC) for fruitful discussions on the geomorphological settings of the study area. This work was also supported by the I-AMICA (PONa3_00363) project. The authors would like to thank European Commission, Copernicus Programme and ESA for the systematic 6-day Sentinel-1 data acquisition and their open data policy. The Digital Elevation Models of the analyzed areas were acquired through the NASA SRTM archive. This article contains modified Copernicus Sentinel data 2022. The authors thank the anonymous reviewers for their criticisms and comments that were useful for improving the manuscript.

**Conflicts of Interest:** The authors declare no conflict of interest.

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
