# Peer review of "A Procedure for the Quantitative Comparison of Rainfall and DInSAR-Based Surface Displacement Time Series in Slow-Moving Landslides: A Case Study in Southern Italy"

_remotesensing, doi:10.3390/rs15020320_

Round 1

Reviewer 1 Report (Previous Reviewer 4)

The authors have revised the paper thoroughly according to my previous comments. I have no more comments.

Reviewer 2 Report (New Reviewer)

I read the manuscript carefully and with great pleasure. Clearly written and in an understandable style. The methodology is well described, the article is well structured. The research was done correctly, without unnecessary interpretations. The study area is well chosen and well described geologically and geomorphologically. The results are clearly presented together with a good discussion. The approach used will have an indisputable application in practice. I have nothing to add except to recommend that the English language be checked once more. I wish success to the authors.

This manuscript is a resubmission of an earlier submission. The following is a list of the peer review reports and author responses from that submission.

Round 1

Reviewer 1 Report

The actual version of this paper improves the original one

Reviewer 2 Report

I do not see any particular improvement since the former submission. 

Reviewer 3 Report

The paper under consideration is devoted to the estimation of the impact of rainfall on the slow‐moving landslides using quantitative procedure for identifying the satellite‐based displacement in comparison with rainfall series. The paper is based on the processing and analysis of huge amount of Sentinel-1A and B SAR data acquired in 2015-2020. The area covered with standard Sentinel-1 IW SLC data is Basilicata region including Basento, Bradano, Agri, and Sinni river catchments. Authors applied P-SBAS approach to process interferometric time series on ascending and descending orbits.

Resulting displacements maps were analyzed and two landslides were selected for a more detailed study of the effect of rain on landslides dynamics.

Although the work done is impressive, there are some remarks and a number of issues that I would like to discuss.

1. The marks of average velocities and amplitudes of cumulative deformations in the figure 4 are practically inconsistent with the location of landslide spots from the inventory of the landslide phenomena in Italy. Why?

2. Authors mentioned “Spectral Diversity procedure” in line 238. That is only one of the stages of Sentinel-1 SAR data processing chain, and not the most important. Say, ideal co-registration could make this procedure unnecessary. I think co-registration is also worth mentioning.

3. Captions of Figs 5 b and 5c are mixed up.

4. Authors are worrying about impact of rainfalls on the displacements. How can they guarantee an absence of other triggering factors during the 5years time interval?

5.Sorry, Fig. 10 f is difficult to understand. Need more explanation.

6. In the Discussion section authors declare that “The procedure proposed in this work allows for the identification of active areas of landslides, by means of the identification of clusters of pixels measuring satellite‐based surface displacements, and for the comparison among series of displacements and rainfall measurements”. Isn't the very fact of movement a sign of the landslide? Or they mean that the fact of some cumulative rainfall before the displacement is an additional guarantee of the landslide type of movement?

7. Another interesting statement (lines 471-473): “Furthermore, ground displacement pixels obtained through P‐SBAS analysis are mostly located along road infrastructure and at buildings in urban areas, where landslides are often absent because they are not reported or, if present, already stabilized by anthropic intervention.” Does that mean that the P-SBAS results are unreliable?

Reviewer 4 Report

A procedure to analyze the corelation between rainfall events and ladslide displacements is proposed in this paper, and a case study is reported as an example for this procedure. The major problem is that, the paper is oversimplied and many processes involved in the procedure are not clearly explained. Also, the effectiveness of the proposed procedure has not been verified. How can we really trust and rely on the analysis results provided in the case study?  

1.     L52: such->such as

2.     L57: due->due to

3.     The DInSAR Analysis section should be improved and the P-SBAS processing chain should be explained in more details, since this is the most important remote sensing technique involved in this study. Using some equations and some schematic diagrams might be helpful.

4.     In the Statistiacal analysis section, there should be some equations to explain the Kendall rand correlation coefficient and the Maximal Information Coefficient.

5.     There should be legends to explain the meanings of all the curves in Figure 8a and Figure 9a.

6.     L448: between rainfall series and relative displacements and rainfall series -> between relative displacements and rainfall series
